# Moral Certainty of the Judge in the Canonical Process to Determine the Nullity of Marriage v. the Principle *Testis Unus Testis Nullus*

Karol Krystian Adamczewski 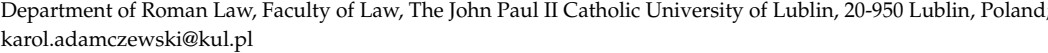

Department of Roman Law, Faculty of Law, The John Paul II Catholic University of Lublin, 20-950 Lublin, Poland; karol.adamczewski@kul.pl

**Abstract:** The present article discusses the issue of moral certainty in the canonical process for the annulment of marriage and analyzes the problem of applying the procedural principle *testis unus testis nullus*. The reason for undertaking the subject was the publication in 2015 of two papal documents of the rank of apostolic letters, which introduced significant changes in the area of the process for the annulment of marriage. One of the aspects that underwent legislative modification was the principle *testis unus testis nullus*. It was decided that in marriage cases the testimony provided by a single witness may have the value of complete proof, provided that certain conditions are met. Therefore, the current considerations are an attempt to find an answer to the question whether the judge who takes a decision in marriage cases on the grounds of a single-witness testimony is able to achieve inner conviction (*certitudo moralis*) with regard to the factual state of the matter, and pass a verdict in accordance with the truth. In addition, the article outlines the legal construct of moral certainty, characteristic of canon law, and discusses the principle *testis unus testis nullus* within the framework of the existing canon law, taking into account a broad historical perspective, including both the biblical-canonical tradition as well as Roman law.

**Keywords:** moral certainty; *testis unus testis nullus*; witness testimonies; process for the annulment of marriage

## 1. Introduction

Exquisite German Romanist Andreas Wacke was of the opinion that preserving and respecting the principle *testis unus testis nullus* resulted from the awareness that it was deeply rooted both in human as well as God's law (Wacke 1997, pp. 50–51). Undoubtedly, it is hard to argue with the above opinion, even more so in view of the fact that it is supported by compelling arguments. Indeed, the requirement of providing at least two witnesses had been laid down not only in imperial Roman law, but also in the Bible. The obligation of obtaining at least two consistent testimonies from witnesses was issued in the form of a constitution by Emperor Constantine the Great in the year 334 after Christ.[1] It is significant that the principle *testis unus testis nullus* was well known and firmly established in both the *Old*[2] and *New Testament*[3] several centuries before. In its essence, the principle prohibited the judge to pass judgement on the grounds of an uncorroborated statement by a single witness. It was required that at least two witnesses should give a consistent testimony in the case. The application of the principle of evidence in question in the proceedings was meant primarily to safeguard the procedural rights of the accused, especially by making it more difficult to bring false statements before the court. Furthermore, the correct application of the requirement increased the likelihood of a just trial and arriving at the truth, which appeared to be a fundamental purpose of legal proceedings (Metro 2001, pp. 110–11). Therefore, it seems natural that the principle *testis unus testis nullus* played a key role in the law of the Church, especially in canonical proceedings and in cases regarding the nullity of marriage, which constitutes one of the special canonical proceedings. However, the

direct reason for addressing the issue in the present perspective was the last reform of the processes regarding the nullity of marriage, of 2015. The conducted modification of the legal processes for the nullity of marriage was effected in the publication of, among others, two apostolic letters, *Motu proprio*: *Mitis Iudex Dominus Iesus*[4] for the Latin Church and *Mitis et misericors Iesus* for the Eastern Churches.[5] The above acts refer back to the ancient rule in the context of marriage-invalidity proceedings, albeit with a significant change in its understanding and application. In both documents the legislator decided, somewhat unexpectedly, that in cases regarding the procedures for declaration of marriage nullity, the testimony of a single witness may have full and complete evidentiary effect if appropriate provisions are met. Nevertheless, it should be observed that the requirement *testimonium unius* under c. 1573 CIC[6] remained in force. In view of the above, it is justifiable to investigate whether the above reform waived the ancient rule in cases for declaring marriage nullity. Or perhaps it has changed to some extent the concept of understanding and applying the requirement of double testimony? Finally, it seems essential to verify the correlation between the obligation for the judge to reach moral certainty in cases for declaring marriage nullity, and the admissibility on certain conditions of evidence provided by a single witness.

## 2. Moral Certainty in Canon Law

In accordance with c. 1608 § 1 Code of Canon Law: "To give any judgement, the judge must have in his mind moral certainty about the matter to be decided in the judgement."[7]. In the paragraph cited above, the legislator indicated that the concept of moral certainty is based on the conviction of the judge concerning the truthfulness or falsity of the given fact or situation (Erlebach 2007, p. 267). Thus, in the context of the process for declaring marriage nullity, which by its very nature is connected with the question of public good, the primary objective is to determine the existing reality, in this case the existence or non-existence of a valid marriage (Mierzejewski 2013, p. 139). As a result, the issue of moral certainty had long been the subject of profound concern from the perspective of the doctrine of canon law, and was the subject of thorough consideration in papal teaching, especially during the pontificate of Pius XII and John Paul II. The significance and role of this issue is very accurately represented in the words of one of the most distinguished scholars from the field of canon law, Edward McCarthy, who stated that the legal construct under discussion constitutes the central point and a synthesis of the canonical procedural law (McCarthy 1948, pp. 60–61).

It seems that moral certainty, which should accompany the Church judge in the process of resolving the subject matter of the dispute, is equivalent to neither metaphysical certainty nor high probability. Neither is it a subjective conviction (De Diego-Lora 2011, p. 1206). In a visual sense, it might be placed between two extremes. Indeed, moral certitude may be situated between absolute (philosophical) certainty, characteristic of philosophical discourse, and absolute quasi-certainty, marked by a high level of probability (Zannoni 2015, pp. 125–26).

It is vital that moral certainty should be based not only on the rules of logic, but also on the ethical principles governing human behaviour. It has been emphasized that sometimes the truth crucial to the court's final decision was not available in the form of direct evidence, but it was arrived at in the course of deductive reasoning, conducted on the basis of objective data. The latter might include, for instance, circumstantial evidence, statements of the parties, and other evidentiary data. In accordance with the papal teaching, which has significantly contributed to the understanding and implementation of the institution of moral certitude, it has been observed that it is not uncommon for arrival at moral certainty to require painstaking analysis of numerous clues and fragments of data. These might not be sufficient when assessed individually, but when combined they constitute an effective mechanism of arriving at the said objective (De Diego-Lora 2011, p. 1206).

The legislator laid down that the appraisal of evidence is conducted by the judge within his conscience.[8] This corresponds to some extent with the rule of judicial discretion.

However, it has been pointed out that occasionally it might be restricted by law through provisions which directly inform the judge about the effectiveness of particular evidence (Grzywacz 1985, vol. 5, p. 35). A perfect example of this type of intervention by the legislator is the procedural construct prohibiting the acceptance of *testimonium unius*. It should be emphasized that the existing Roman legal principle *testis unus testis nullus*, contained, among others, in c. 1753 CIC, can boast an impressive period of being in force through its application in both the Judeo-Christian and Roman traditions.

By identifying the source of moral certainty, the Church legislator indicated that, in accordance with c. 1608 § 2 CIC: "The judge must derive this certainty from the acts of the case and from the proofs". According to Zenon Grocholewski, the institution of moral certainty provides a clear criterion for the judge as to what state of mind he should be in while determining the case, taking into account, necessarily, the collected *acta et probata*. In other words, it establishes a substantial benchmark which takes into consideration the limits of the human mind and, at the same time, respects and demands the truth. It is a criterion reflecting prudence and rationality (Grocholewski 1998, pp. 22–23).

It should be observed that the certitude in question does not have to be absolute, that is, it does not exclude the element of the doubt and the possibility of making an error. The condition for the existence of *moralis certitudo* will be met if at least any reasonable doubt and justified degree of uncertainty are excluded. In practical terms, it means that the judge taking a decision cannot base his judgement merely on conjecture or even a state of high probability. What is required is ruling on the grounds of the existing data pertaining to the state described as objective moral certainty. Therefore, a judge's conviction highly subjective in nature, based for example on his specific preferences or private opinions, should be rejected (Pawluk 1990, pp. 290–91).

Andrea Bettetini accurately pointed out that the activity of the judge in the procedure does not entail a "sterile" and neutral approach aimed at recreating the necessary and significant facts from the perspective of procedural objectives. The process and engagement of the judge cannot be based on merely the conducting of a "procedural arithmetic operation" with the precision and linearity equaling perfection. On the contrary, he argued that in a natural way in each case assessed by the judge there exist varying degrees of his involvement with regard to the parties and witnesses concerned. In consequence, it is impossible for the judge to maintain complete neutrality. The Italian scholar was also of the opinion that it was by no means a coincidence that c. 1608 § 3 CIC, being a confirmation of the centuries-old normative tradition in this respect, states that the judge must weigh the facts and evidence in his conscience (*ex sua conscienta*) and, as further stated in c.1608 § 4 CIC, when the judge cannot arrive at such certitude (*se eam certitudinem adipisci non potiut*), he should not pass a positive ruling, that is in favour of the plaintiff's claim (Bettetini 2019, pp. 58–59).

Undoubtedly, a huge impact on the creation and shaping of the original concept of moral certainty in canon law, corresponding to the principle *testis unus testis nullus*, was the thinking of St. Thomas Aquinas, based on the teaching of Aristotle (Zannoni 2015, pp. 184–85).

Aquinas believed that the number of witnesses is a significant element that has influence on the process of assessing the case carried out by the judge. He reasonably argued that a large number of witnesses does not in itself guarantee the accuracy of the testimonies, nor does it allow the judge to reach a state of moral certainty. However, he emphasized that consistent testimonies from two or three witnesses constitute in themselves significant value and bring the judge closer to a state that might be described as "more probable certainty". Hence, aspects of the number and consistency of the statements made before the court must be taken into consideration (Tomasz z Akwinu 2006, pp. 57–80, transl. pp. 218–19).

While considering the issue of certitude in the legal process, Aquinas emphasized that it is not necessary to seek the same certainty in everything. Referring to works by Aristotle, he indicated that the statements of the witnesses as well as the rulings of the

judges, the latter based on the former, cannot reach the value of absolute certainty based on strict proof (*certitudo demonstrativa*), because they belong to the category of human actions, which by their very nature are of accidental and changeable character (*Ibidem*, p. 218). It is for this reason that Aquinas accepted probable certainty (*probabilis certitudo*), explaining that in most cases it equals the truth, yet acknowledging also that sometimes it might fail (Bettetini 2019, pp. 56–57). Aquinas further emphasized that a consistent testimony from two persons is closer to the truth than one from a single witness. Pointing to the crucial importance of the number of witnesses in arriving at the truth, Aquinas noted that the requirement in the given case of two or three witnesses providing a consistent testimony embodies a perfect multitude (*multitudo perfecta*) and thus provides warranty for an increased certainty of reaching the objective under discussion (Tomasz z Akwinu 2006, p. 218).

### 3. "Moral Certainty" in Roman Law

Undoubtedly, the legal construct of moral certainty constitutes an achievement of the doctrine of canon law, based to a large degree on Christian philosophy and moral theology. Nevertheless, the influence of Roman law in this matter cannot be excluded from discussion. A specific type of *exemplum* that might confirm the above-mentioned hypothesis is the text contained in the Digest of Justinian, being a fragment of Emperor Hadrian's constitution (76–138 AD), addressed to Valerius Verrus, a Roman official in Cilicia. In the imperial rescript, the emperor conveyed important instructions to his subordinate with regard to collecting and evaluating evidence in the legal process, especially with respect to the assessment of the credibility of witnesses. The emperor explained to his associate that the judge's conviction (*animus*), among other factors, is always crucial and decisive in finding a legal resolution. The decision should be made after a conscientious observation of facts and appraisal of evidence, with the inclusion of numerous circumstances which contribute to the explanation of the whole matter and which must be distinctly marked. Hadrian wrote to his official in Cilicia in the following words:

[I]t cannot be defined in advance in a sufficiently certain way what evidence and in what amount is sufficient to prove each individual case. It frequently happens, even though not always, that the truth in a given case is determined without resorting to public records. Sometimes the credibility of the circumstances which are subject to investigation is supported by the number of witnesses, sometimes by the dignity and authority that the witnesses enjoy and sometimes by common and popular opinion. Therefore, I can tell you in the rescript in a general way only this that under no circumstances should the assessment of the case be limited only to a single type of proof but it is necessary that you make a judgement on the basis of your inner conviction as to what you believe [to be true] or what has not been sufficiently proven.[9]

The expression *animus* used in the text has several meanings. It may be understood as life spirit, soul, mind, true conviction, or even heart (Plezia 1959, pp. 199–201). On the other hand, the word *conscientia*, used in c. 1608 § 3 CIC translates as, among other definitions, knowledge shared with others, awareness, knowledge of something, conscience (*Ibidem*). In light of the above considerations, it seems that the fragment appearing in the Digest: *sed ex sententia animi tui te aestimare oportere*[10] together with the expressions from the Code used by the canonical legislator, especially *requritur in iudicis animo moralis certitudo*[11] and *probationem autem aestimare iudex ex sua conscientia*[12], oscillate around the same essential issue, that is, human spirit or human conscience and the possibility of learning the truth, even though they are not the same in their literal meanings.

It might be therefore assumed that the problem of the accurate inner attitude of the judge and that of the well-formed conscience (*conscientia*) were also the subject of imperial interest in the era of the Dominate. The topic was raised in one of the constitutions of Constantine the Great of 333, referred to as the *First Sirmondian Constitution*, which regulated, apart from other things, the issue of *episcopalis audientia*[13] and the position of the Catholic clergy in resolving secular matters (Adamczewski 2022, p. 156).[14] It regulated,

among other matters, the question of the authority of the bishop exercising the judicial powers on behalf of the emperor, as well as the value and power of his testimony with regard to the office he held. While giving specific instructions to his subordinate Falvius Ablabius, a praetorian prefect in Constantinople, the emperor wrote the following:

The testimony provided even by a single bishop should be accepted by each judge without any doubt . . . Indeed, such a testimony is strengthened by the authority of the truth and is indisputable as it was issued by a consecrated man of immaculate conscience.[15]

It is possible that Constantine the Great wanted to underscore in this way the importance of the relation between the bishop's inner attitude (conscience), described as *sacrosancto homine conscientia mentis*, and his testimony, delivered in public within the frame of exercising *episcopalis audientia*, as a certain guarantee of fair judgement that would be consistent with the truth.

### 4. The Application of the Principle *Testis Unus Testis Nullus*

In his considerations on moral certainty regulated by the provisions of a previous codex, Władysław Szafrański pointed out that the judge should give a definitive verdict on the whole of the matter only after he has thoroughly considered all circumstances speaking in favor of and against a given thesis. He argued that moral certainty in such a case would be the result of the judge's prudence. He also warned of various situations in which the judge would not be able to reach moral certainty in marriage cases. In such situations, the marriage should be declared valid. The obligation to follow such a procedure transpired directly from canon law, which contains the principle regulating the essence of the validity of marriage, i.e., *standum est pro valore matrimonii*. The above-mentioned author observed that in some circumstances the Church legislator set much less rigorous requirements. As an example, he indicated c. 1791 § 2 CIC[16]. Interestingly, he suggested, the disposition contained in the cited canon was a manifestation of a "milder" attitude on the part of the legislator, who allowed the judge to act effectively upon the sworn testimonies of two or three credible witnesses on condition that they testified *de scientia de aliqua re vel facto* (Szafrański 1958, p. 303–4).

A case in point in view of the above deliberations is a 1942 statement of Pius XII, addressed to the workers the Roman Rota. The Bishop of Rome observed that in a situation when, on the basis of the collected evidence, where each piece of evidence was appraised individually, it was not possible to deem any of them sufficient, then it was necessary to take a synthetic assessment of the whole. If the synthetic assessment of the collected material and a broader view changed the judge's perspective and led him to the state of moral certainty, it would be sufficient for making an accurate judgement. Jerzy Grzywacz suggested that the papal teaching in this matter, being a long continuation of Church jurisprudence, in practice meant an allowance to seek resolution in marriage cases on the basis of a single testimony. As the author noted, even though the statement did not include *expressis verbis* a reference to the disposition under c. 1791 § 1 CIC, still the tone and context of the papal interpretation in a clear way gave permission for making use of the said canon and for accepting testimony from a single witness (Grzywacz 1985, pp. 36–37). Moving further, the author admitted that the judge would not be able to waive the application of the principle *testis unus testis nullus* if he had no access to "supplement information or evidentiary support" which might be conducive to creating a kind of synthetic assessment of the whole, as specified in the papal teaching. It is worth adding that this issue was to some extent solved in the Code of John Paul II on the grounds of the construction of canon 1753 CIC, which introduced a second exception to the rule in the following wording: "the circumstances regarding things or persons suggest otherwise". In his detailed analysis of the rulings of the Rota issued over the course of the past century, the above-mentioned canonist noted that before the Code of John Paul II was announced, there had appeared a clearly observable tendency in the case law of the Rota that proof from the testimony of a single witness in marriage cases was becoming not only conspicuous, but also increasingly acceptable. However, it was always treated as an exception to the rule and all the other

conditions had to be complied with. The author added that the roots of the trend reached as far back as the 17th century. Since that time, there had been situations where the decisions of the Rota were based on the testimony of only one witness and deemed sufficient to adjudicate in marriage cases. At the same time, the practice was not considered to be contrary to the disposition of c. 1791 § 1 CIC (Grzywacz 1985, p. 50).[17]

It seems that attitudes towards this issue were gradually evolving towards relaxing the prohibition of a single witness on the grounds of the above-mentioned papal reform. The reformed c. 1678 § 2 MIDI allowed for a situation in a marriage case in which establishing the decisive factor was based on the testimony of a single credible witness. What is more, in that case the said evidence had to be accepted since if it had been questioned as *probatio plena*, it might have resulted even in an allegation of perpetrating grave injustice by the judicial authorities (Rozkrut 2015, pp. 109–10). Therefore, it seems that compliance with the minimal normative requirements provided for in the cited canon is key for proper judicial proceedings. In other words, one testimony from a witness and "the circumstances regarding things or persons" should be assessed together.

While analyzing this problem, Iwan Milotić observed that in practical terms there are two conditions to be met. First of all, there must not be any shadow of a doubt or uncertainty with regard to the witness or his credibility. Hence, the judge hearing a witness should be able to make up his mind during the course of the hearing as to the witness's credibility, having acquired the necessary knowledge in this regard beforehand. This is crucial since if the judge has not reached proper certainty or if he merely has doubts regarding the person then, as a rule, the testimony of a single witness should not be accepted.

The second condition refers to the necessity of adjudicating in the state of moral certainty. In other words, if a judge on the basis of a statement of only one witness acquired certainty as to the existence of decisive evidence and if he deemed that, on the basis of all the circumstances of the case, it would create a clear injustice if he rejected the testimony merely to respect the principle *testis units testis nullus*, then he should accept the testimony from a single witness as complete evidence (Milotić 2019, pp. 856–57).

## 5. Conclusions

It is obvious that in light of the above arguments and other essential changes (Adamowicz 2015, pp. 67–74)[18] implemented within the frame of the code reform of 2015, the role and responsibility of the judge increased further. As it transpires, it is expected of the judge to demonstrate a prudent and cautious attitude, as well as considerable experience, allowing him to properly assess the situation in which there are reasons for accepting the testimony from a single witness. It also presupposes that the judge should adopt an even more active role in the process, as the legislator left the final interpretation of each individual case and the value of a single testimony, combined with the "circumstances regarding persons or things", within the competence of the judge (Milotić 2019, p. 857).

Two papal documents of the rank of apostolic letters, published in 2015, introduced serious changes in the process for declaring the annulment of the marriage. One of the legislative modifications was the tool prohibiting, as a general rule, the acceptance of testimony from a single witness as complete proof. Indeed, it was decided that in marriage cases the testimony of only one witness may be considered as complete proof, provided that certain specific conditions are met. In fact, the doctrine takes the stance that the reform has not devalued the importance of the classic principle *testis unus testis nullus*. Nevertheless, it seems that the legislative initiative opened the way to or was the next step towards a more permissive approach from the judge who has at his disposal only testimony from a single witness. Still, even the most liberal understanding and attitude to the problem, as a consequence of the reform of c. 1678 § 2 MID, should be by no means regarded as the establishment of a new rule. It might be hoped that the intention of the legislator was precisely to indicate an exception to the strict adherence to the rule (*Ibidem*).

### 6. Final Considerations

In the context of the analyzed issue, it is worth paying attention to the significant regularity in the shaping of its interpretation. Announcing the principle *testis unus testis nullus* as the law commonly in force, Constantine the Great was undoubtedly under the influence of the Judeo-Christian tradition, where double testimony was demanded. Additionally, according to other opinions, he simply confirmed with his imperial authority the policy that was already in common use in case law. In other words, already in the times of the Republic, judges treated single-witness testimony with a great deal of mistrust. Therefore, the rule that was commonly practised and well established in the area of Roman case law was elevated to the status of a commonly binding norm. Over the course of the last century, Church law has been faced with a tendency to the contrary. This has become explicit especially in the aforementioned reform, which under certain conditions allowed testimony from only one witness as sufficient evidence for declaring the annulment of marriage. It might be assumed that the intervention of the Church legislator was in fact a confirmation of the existing tendency in Church jurisprudence and case law, which, despite the established prohibition of single testimony, in duly justified cases deviated from its general enforcement. However, it should be recognized that, contrary to the Constantine's intervention of 334, aimed at strengthening the principle within the system of law, the reform of 2015, inspired to some extent also by case law in the courts, constituted a turn towards legislative liberalization of the principle *testis unus testis nullus*.

It is noteworthy that one of the motifs of granting greater value to single-witness testimony or statements made by the parties was to ensure their sufficient appreciation. It was also intended to be a demonstration of greater trust in such testimonies, basing it on the condition that the parties giving such testimonies do intend to contribute to establishing the truth, even though for centuries the legislator had been genuinely aware of human frailty and, unfortunately, constant propensity for lying.

The principle *testis unus testis nullus* in its essence expressed considerable distrust in a situation where a deciding factor was established on the basis of a single-witness testimony as an only means of proof. It might seem that the eagerness of the Church legislator to make court rulings based on evidence *luce meridiana clarioribus*, which had been observable for centuries, was changed to some degree, especially in cases regarding the annulment of marriage.

**Funding:** This research received no external funding.

**Data Availability Statement:** Not applicable.

**Conflicts of Interest:** The author declares no conflict of interest.

### Notes

[1] See: Th. 11.39.3; C. 4.20.9.

[2] See: Deut 19, 15; Deut 17, 6–7; Num 35,30; Dan 13, 1–64; 1 Kings 21, 1–15.

[3] See: Mt 18, 19–20; Lk 10, 1; Jn 8, 17; Jn 18, 21; Mt 18, 16; Mt 26, 60–65; Mk 14, 55–59; Lk 22, 71; Mt 17, 1–3; 2 Cor 13, 1; 1 Tim 5, 19–20; Heb 10, 25–29; Acts 6, 13; Apoc 11, 3–4; 7–8.

[4] See: 1678 § 2 KPK: In the same cases, the testimony of one witness can produce full proof if it concerns a qualified witness making a deposition concerning matters done *ex officio*, or unless the circumstances of things and persons suggest it. Papież Franciszek, *List apostolski motu proprio Mitis Iudex Dominus Iesus, reformujący kanony Kodeksu Prawa Kanonicznego dotyczące spraw o orzeczenie nieważności małżeństwa*, tekst łacińsko-polski, Tarnów 2015.

[5] See: 1364 § 2 CCEO: In the same cases, the testimony of one witness can produce full proof if it concerns a qualified witness making a deposition concerning matters done *ex officio*, or unless the circumstances of things and persons suggest it. Papież Franciszek, *List apostolski motu proprio Mitis et misericors Iesus, reformujący kanony Kodeksu Prawa Kanonicznego dotyczące spraw o orzeczenie nieważności małżeństwa*, tekst łacińsko-polski, Tarnów 2015.

[6] See: c. 1573: The testimony of one witness cannot produce full proof unless it concerns a qualified witness making a deposition concerning matters done ex officio, or unless the circumstances of things and persons suggest otherwise. *Codex Iuris Canonici*, 25 I 1983, AAS 75 (1983), pars II, pp. 1–317 with subsequent changes; *Kodeks Prawa Kanonicznego*, current legal solutions as of 18 May 2022, updated translation into the Polish language, Poznań 2022.

[7]　The legal construct of moral certainty was regulated in an analogous way in the law of the Eastern Churches. See: c. 1291 CCEO.

[8]　C. 1608 CIC: "§ 3. The judge must conscientiously weigh the evidence, with due regard for the provisions of law about the efficacy of certain evidence. § 4. A judge who cannot arrive at such certainty is to pronounce that the right of the plaintiff is not established and is to find for the respondent except in a case which enjoys the favour of law, when he is to pronounce in its favour.".

[9]　D. 22.5.3.2: *Eiusdem quoque principis exstat rescriptum ad Valerium Verum de excutienda fide testium in haec verba: Quae argumenta ad quem modum probandae cuique rei sufficiant, nullo certo modo satis definiri potest. Sicut non semper, ita saepe sine publicis monumentis cuiusque rei veritas deprehenditur. Alias numerus testium, alias dignitas et auctoritas, alias veluti consentiens fama confirmat rei de qua quaeritur fidem. Hoc ergo solum tibi rescribere possum summatim non utique ad unam probationis speciem cognitionem statim alligari debere, sed ex sententia animi tui te aestimare oportere, quid aut credas aut parum probatum tibi opinaris.* Transl. under the editorship of T. Palmirski.

[10]　"So that you could judge in accordance with your inner conviction," see: D. 22.5.3.2.

[11]　"The judge must conscientiously weigh the evidence," see. c. 1608 § 3 CIC.

[12]　" . . . the judge must have in his mind moral certainty," see: c. 1608 § 1 CIC.

[13]　*Episcopalis audientia* was the right granted to bishops by the imperial power (the beginning of the 4th c.) for the purpose of executing the judicial authority also in cases regarding lay persons. It was one of the first Church institutions which was formally acknowledged by Roman law and enjoyed an extensive appreciation among Roman society. See: (Cimma 1989, p. 5) and next.

[14]　For more information on the subject see: (Adamczewski 2022, p. 156) and next.

[15]　Const. Sirm. 1: *Testimonium etiam ab uno licet episcopo perhibitum omnis iudex indubitanter accipiat ( . . . ). Illud est enim veritatis auctoritate firmatum, illud incorruptum, quod a sacrosancto homine conscientia mentis illibatae protulerit.*

[16]　C. 1791 § 2 CIC: "A sufficient proof may be provided in the testimony of two or three credible witnesses who under oath and to their best knowledge will testify before the court in a completely consistent way about the same thing or the same fact. In some very important matters or in case of circumstances raising doubt as to whether the truth of the statements has been sufficiently demonstrated, the judge may require a more complete proof.".

[17]　*Ibidem*, p. 50. The said author enumerates in detail and analyzes the rulings of the Roman Rota Tribunal, especially those which have been issued on the basis of a testimony from a single witness, delivered in compliance with all the relevant requirements.

[18]　One of the essential changes introduced under the *motu proprio Mitis Iudex Dominus Iesus* of 2015 is the abolishing of two consistent court decisions in marriage cases. The legislator offered similar arguments in favour of this modification, among others, the need to simplify the marriage process in the appeal proceedings. An interesting analysis with regard to this issue, including the arguments in favour of and against the implemented reform, has been offered by a Polish canonist Leszek Adamowicz. See (Adamowicz 2015, pp. 67–64).

## References

### Archives Sources

*Pismo Święte Starego i Nowego Testamentu*, w przekł. z języków oryginalnych. In *Biblia Tysiąclecia*. Edited by A. Jankowski, A. Stachowiak and K. Romaniuk. Poznań—Warszawa 1980.

*Codex Iuris Canonci Pii X Pontificis Maximi iussu digestus Benedicti Papae XV auctoritate promulgatus, in:* AAS (1917-II), pp. 5–521.

*Codex Iuris Canonici auctoritate Ioannis Pauli pp. II promulgates (25.01.1983)*, AAS 75 (1983), pars II, pp. 1–317 with subsequent changes; *Kodeks Prawa Kanonicznego*, current legal solutions as of 18 May 2022, updated translation into the Polish language, Poznań 2022.

*Codex Canonum Ecclesiarum auctoritate Joannis Paulii PP. II promulgatus, AAS 82 (1990)*, no. 11, pp. 1033–1353; *Kodeks Kanonów Kościołów Wschodnich, promulgowany przez papieża Jana Pawła II*, Translation, preface to the Polish edition, preparation of the source material, index, glossary and electronic edition L. Adamowicz, transl. M. Dyjakowska, Lublin 2001.

Franciscus PP., *Litterae apostolicae motu proprio datae Mitis Iudex Dominus Iesus quibuscanones Codicis Iuris Canonici de Causis ad Matrimonii nullitatem declarandam reformatur* (15.08.2015), AAS 107 (2015), s. 958-970; tekst polski w: Papież Franciszek, *List apostolski motu proprio Mitis Iudex Dominus Iesus reformujący kanony Kodeksu Prawa Kanonicznego dotyczące spraw o orzeczenie nieważności małżeństwa*, tekst łacińsko-polski, Tarnów 2015.

*Theodosiani libri XVI cum Constitonibus Sirmondianis et Leges novellae ad Theodosianum pertinentes Consilio at auctoritatae Academiae litterarum regiae borussicae*, ed. Th. Mommsen, P.M. Meyer, J. Sirmond, Berolini 1905.

*Digesta, w: Corpus Iuris Civilis*, vol. I, ed. Th. Mommsen, P. Krüger, Berolini 1954.

*Digesta Iustiniani. Digesta Justyniańskie. Tekst i przekład*. IV, Księgi 20–27, ed. T. Palmirski, Kraków 2014.

*Codex Iustinianus, w: Corpus Iuris Civilis*, vol. II, ed. P. Krüger, Berolini 1954.

### Published Sources

Adamczewski, Karol Kristian. 2022. *Biblijne i Rzymskie Korzenie Procesowej Zasady Testis unus Testis Nullus*. Lublin: Wydawnictwo KUL.

Adamowicz, Leszek. 2015. Jedna czy dwie instancje?—w oczekiwaniu na reformę procesu małżeńskiego. In *Procesy i procedury: Nowe wyzwania. Materiały z Konferencji Naukowej zorganizowanej na Wydziale Prawa Kanonicznego w dniu 22 października 2015 r*. Edited by Grzegorz Leszczyński. Warszawa: Wydawnictwo Uniwersytetu Kardynała Stefana Wyszyńskiego, pp. 67–74.

Bettetini, Andrea. 2019. *Iustitia et Fides. Studi di diritto canonico processuale e matrimoniale*. Torino: G. Giappichelli.

Cimma, Maria Rosa. 1989. *L'Episcopalis audientia nelle costituzioni imperiali da Costantino a Giustiniano*. Torino: G. Giappichelli.

De Diego-Lora, Carmelo. 2011. Orzeczenie sędziego. In *Codex Iuris Canonici. Kodeks Prawa Kanonicznego. Komentarz. Powszechne i partykularne ustawodawstwo Kościoła katolickiego. Podstawowe akty polskiego prawa wyznaniowego*. Edited by Piotr Majer. Kraków: Wolters Kluwer, pp. 1204–14.

Erlebach, Grzegorz. 2007. Niektóre procesy specjalne. In *Komentarz do Kodeksu Prawa Kanonicznego*. Księga VII. Procesy, ed. J. Krukowski. Poznań: Pallottinum, vol. V.

Grocholewski, Zenon. 1998. Pewność moralna, jako klucz do lektury norm procesowych. *Ius Matrimoniale* 3: 22–23. [CrossRef] [PubMed]

Grzywacz, Jerzy. 1985. Moc dowodowa zeznań świadków według nowego Kodeksu Prawa Kanonicznego. *Roczniki Teologiczno-Kanoniczne* 32: 19–53.

McCarthy, Edward Anthony. 1948. *De certitudine morali quae in iudicis animo ad sententiae pronuntiationem requiritur*. Romae: Officium Libri Catholici.

Metro, Antonino. 2001. Testis unus testis nullus. In *Critical Studies in Ancient Law, Comparative Law and Legal History*. Essays in Honour of Alan Watson. Edited by John W. Cairns and Olivia F. Robinson. Oxford and Portland: Hart Publishing, pp. 109–116.

Mierzejewski, Krzysztof. 2013. Poszukiwanie prawdy obiektywnej a pewność moralna sędziego w kanonicznym procesie małżeńskim. *Prawo Kanoniczne* 56: 137–52. [CrossRef]

Milotić, Ivan. 2019. Testis unus testis nullus u rimsko-kanonskom i važećem kanonskom postupku. *Bogoslovska Smotra* 89: 837–59.

Pawluk, Tadeusz. 1990. *Prawo kanoniczne według Kodeksu Jana Pawła II*. Doczesne dobra Kościoła. Sankcje w Kościele. Procesy. Olsztyn: Warmińskie Wydawnictwo Diecezjalne, vol. IV.

Rozkrut, Tomasz. 2015. Komentarz do kan. 1675–1678 MDI, art. 10–11 Ratio. In *Praktyczny komentarz do Listu apostolskiego motu proprio Mitis Iudex Dominus Iesus papieża Franciszka*. Edited by Piotr Skonieczny. Tarnów: Wydawnictwo Diecezji Tarnowskiej Biblos, pp. 89–121.

*Słownik łacińsko-polski*. 1959. Plezia, Marian, ed. Warszawa: Państwowe Wydawnictwo Naukowe, vol. I.

Szafrański, Władysław. 1958. Pewność moralna w kościelnym wyroku sądowym. *Prawo Kanoniczne* 1: 281–307. [CrossRef]

Tomasz z Akwinu. 2006. *Traktat o sprawiedliwości, Summa teologii II-II*. q. 57–80. Translated and Edited by W. Galewicz. Kęty: Wydawnictwo Antyk.

Wacke, Andreas. 1997. Unus Testis, Nullus Testis. Entstehung und Überwindung des Dogmas vom Legalen Beweismaß. *Fundamina* 3: 49–57.

Zannoni, Giorgio. 2015. *Evento Coniugale e Certezza Morale del Giudice*. L'Interpretazione, Vitale"Della Norma. Studi Giuridici CXVI. Vaticano: Libreria Editrice Vaticana.

