# Peer review of "Moral Certainty of the Judge in the Canonical Process to Determine the Nullity of Marriage v. the Principle Testis Unus Testis Nullus"

_religions, doi:10.3390/rel14030405_

Round 1

Reviewer 1 Report

Today, Roman law lives not only in some private laws, but also in the more than 2-year-old legal tradition of the Catholic Church. Due to the decreasing knowledge of canon law, this phenomenon is increasingly marginalized. Meanwhile, the principle of Ecclesia vivit lege Romana is still valid, which is perfectly shown in the presented article. The spirit of Roman law (a concept formulated by Rudolf von Jhering) lives not only in civil codes, but also in the procedures of the Catholic Church.

I rate the reviewed article very highly. The author does not repeat a slogan, but tries to carry out his own proof that Roman law is still the basis for thinking and acting in the Catholic Church. Canonical procedures have taken a great deal from the Roman cognitive process. As for the axiology of judging, moral theology first forged its own standards. However, it too relied on the thoughts of ancient jurists. I know of no other text in which "moral certainty of judgment" is linked to any ancient original. 

Author Response

Thank you very much for your opinion and suggestions.

Reviewer 2 Report

The author proves to be a canonist of the Catholic Church well inured both with the Roman law and Latin canon law. Then, I remarked the fact that the author had the courageous attitude in tackling a such subject, that questioned even the two papal documents of 2015, which indeed introduced „significant changes in the area of the process 8 for the annulment of marriage”. However, – for the sake of his respect towards the papal authority, the author was obliged to arrive to a happy conclusion: the Apostolical Letters „has not devalued the importance of the classic principle testis unus testis nullus” (sic). Hence my collegial advice: the author has to be more categorical in the choice of his critical scientific attitude, since „amicus Plato, sed magis amica veritas”.

Author Response

Thank you very much for your opinion and remarks. I will try to follow the suggestion.

Reviewer 3 Report

I do congratulate the author both for his laborious juridical and canonical documentation, and for his endeavors to conciliate the two altitudes existing actually in the Catholic Church regarding the canonical procedure of the nullity of marriage.

With your permission, I would like to make some remarks:

a)       We can't speak about „de testimoniorum fide” (cf. can. 1572 and 1573) without to make reference both to the text of the canons and to the canonical doctrine regarding the prohibition of a single testimony.

b)      I advise the author not to remain tributary to the different interpretations accredited by the literature of specialty, in order to conciliate – by all means – the different opinions about the approached subject.  So, „Fontes” first!

c)       The author gives the impression that he endeavored to conciliate the principle „testis unustestis nullus”, deeply rooted both in the divine and natural Law, as well as in the canon Law, with the text of the decisions of the Apostolical Letters of 2015, even though himself realized that a single witness testimony produced „a significant change” in the „understanding and application” of the ancient juridical and canonical principle.

d)      It should be better if the author would present the full content of the two Apostolical Letters of 2015, that the reader does not have the impression that he is „homo unius libri”, since it seems that he used the text of the two papal encyclicals only for the sake of the conciliation of the canons with the papal decisions, albeit  – as it is well known – in the Catholic Church their authority has the force of „jus cogens”, and, consequently, a priority towards the canons.

Despite of these critical remarks, I do propose the publication of the paper after minor revision. If the author takes also into account these remarks, his paper could indeed become a real reference in the literature of specialty.

I do congratulate the author both for his laborious juridical and canonical documentation, and for his endeavors to conciliate the two altitudes existing actually in the Catholic Church regarding the canonical procedure of the nullity of marriage.

With your permission, I would like to make some remarks:

a)       We can't speak about „de testimoniorum fide” (cf. can. 1572 and 1573) without to make reference both to the text of the canons and to the canonical doctrine regarding the prohibition of a single testimony.

b)      I advise the author not to remain tributary to the different interpretations accredited by the literature of specialty, in order to conciliate – by all means – the different opinions about the approached subject.  So, „Fontes” first!

c)       The author gives the impression that he endeavored to conciliate the principle „testis unustestis nullus”, deeply rooted both in the divine and natural Law, as well as in the canon Law, with the text of the decisions of the Apostolical Letters of 2015, even though himself realized that a single witness testimony produced „a significant change” in the „understanding and application” of the ancient juridical and canonical principle.

d)      It should be better if the author would present the full content of the two Apostolical Letters of 2015, that the reader does not have the impression that he is „homo unius libri”, since it seems that he used the text of the two papal encyclicals only for the sake of the conciliation of the canons with the papal decisions, albeit  – as it is well known – in the Catholic Church their authority has the force of „jus cogens”, and, consequently, a priority towards the canons.

Despite of these critical remarks, I do propose the publication of the paper after minor revision. If the author takes also into account these remarks, his paper could indeed become a real reference in the literature of specialty.

Author Response

Thank you very much for your opinion and valuable remarks.
